# Chemiluminescent Receptor Binding Assay for Ciguatoxins and Brevetoxins Using Acridinium Brevetoxin-B2

**DOI:** 10.3390/toxins11100580

**Published:** 2019-10-09

**Authors:** Kazuya Murata, Takeshi Yasumoto

**Affiliations:** 1Faculty of Pharmacy, Kindai University, Higashiosaka 577-8502, Japan; 2Japan Food Research Laboratories, Tama Laboratory, Nagayama 6-11-10, Tama 206-0025, Japan; yasumotot@jfrl.or.jp

**Keywords:** ciguatera, ciguatoxins, neurotoxic shellfish poisoning, brevetoxins, acridinium ligand, chemiluminescent binding assay, synaptosome

## Abstract

Ciguatera is the term for poisoning resulting from eating fish from tropical or subtropical regions. The causative toxins collectively named ciguatoxins (CTXs) widely differ in structures depending on their geographic origins, which range from the Pacific Ocean and the Indian Ocean to the Caribbean Sea. Neurotoxic shellfish poisoning (NSP) is caused by the ingestion of bivalve shellfish contaminated with brevetoxins (BTXs). Structurally, both CTXs and BTXs consist of fused ether rings aligned in a ladder shape. Pharmacologically, they bind at the same site (site-5) of voltage-gated sodium channels. However, the great structural diversity and the rare availability of reference toxins hinder LC-MS and ELISA methods, which operate on structure-based recognition. In this study, we prepared a chemiluminescent ligand, acridinium BTXB2 (ABTX), and tested its suitability for use in competitive binding assays to detect CTXs and BTXs. The affinity of ABTX to the rat brain synaptosome estimated by *K*_i_ (1.66 pM) was approximately two-fold higher than that of PbTx-3 (BTX3). In addition, the equilibrium dissociation constant (*K*_D_) was 0.84 nM, the maximum number of binding was 6.76 pmol toxin/mg protein, and the detection limit was 1.4 amol. The assays performed on samples spiked with CTX3C or BTXB4 (*N*-palmitoylBTXB2) at 0.2–1.0 ng CTX/g fish flesh, and 200–800 ng BTXB4/g shellfish showed a linear relationship between the theoretical and observed toxin amounts.

## 1. Introduction

Ciguatera or Ciguatera fish poisoning (CFP) is a type of seafood poisoning occurring in tropical and subtropical areas. The annual number of patients affected by CFP is estimated to be around 50,000 [1]. Its symptoms involve gastrointestinal and neurological disorders characterized by a reversed perception of temperature. In severe cases, suffering may last for months to years. The cause of CFP is attributed to ciguatoxins (CTXs). In the Pacific, the toxins originate from benthic dinoflagellates, *Gambierdiscus* spp. [2]. Two types of toxins differing in the backbone skeletons occur in the Pacific: ciguatoxin-1B type and ciguatoxin-3C type. While moving up the food chain from herbivorous to carnivorous fish, the toxins are oxidized by the CYP enzyme to as many as 24 congeners [3,4,5]. In the Caribbean Sea, different skeletal structures such as C-CTX1 and C-CTX2 are known, but the concurrent toxins remain unknown. Also unknown are the structures of CTXs occurring in the Indian Ocean (I-CTX) [6]. The diverse or unknown structures are the serious obstacles to applying LC-MS [7] and ELISA [8]. Similar problems are encountered in neurotoxic shellfish poisoning (NSP). The causative toxins called brevetoxins (BTXs), or PbTx in some publications, are produced by planktonic dinoflagellates namely, *Karenia* spp., which accumulate in shellfish, and causes intoxication upon ingestion of BTXs. BTXs resemble CTXs in possessing a ladder-shaped polycyclic ether skeleton and binding the same site (site-5) of the voltage-gated sodium channel (Navch) [9,10,11,12,13]. Similar to CTXs, BTXs originally produced by the dinoflagellates undergo metabolic changes in shellfish, and so the number of active metabolites is difficult to analyze [14]. Under such circumstances, a function-based assay seems to be a favorable choice, because it operates regardless of individual structures. One of the metabolites in shellfish, BTXB2 [11], possesses an amino group available for labeling with a marker moiety. We chose a chemiluminescent moiety for labeling, because background interference was expected to be lower than that of the fluorescent moiety. Acridinium-BTXB2 (ABTX, Figure 1) thus prepared showed promising properties. The binding of ABTX to rat brain synaptosome was two times stronger than that of brevetoxin-3, the major BTX produced by the dinoflagellate. In addition, a detection limit as low as 1.4 amol for ABTX was achieved.

## 2. Results

### 2.1. Preparation of ABTX

From 200 µg of BTXB2, 59 µg of ABTX was obtained, and the overall yield was 22%. The low yield can be attributed to the use of water (buffer solution) in the reaction mixture. In the fast atom bombardment (FAB) MS spectrum of ABTX, the molecular ion peak ([M+H]^+^) was observed at *m/z* 1401, and was selected as a precursor ion to perform collision-induced dissociation (CID)-FAB MS/MS. The spectrum and the assignment of fragment ions are shown in Figure 2. In CID-FAB MS/MS spectrum, the peaks indicating the fragments were derived from degradation of acridine moiety (*m/z* 193, 221, 238, 313, 340, 368), which were clearly observed, thereby confirming the molecular structure.

During chemiluminescent measurements of ABTX, the intensity rose immediately after the addition of the trigger solution, and decreased to the basal level within 10 seconds (Figure 3). From this result, the integration time was set to 50 seconds. From optimized measurement conditions, the linearity of ABTX was confirmed from the calibration curve (Figure 4), and ABTX could be detected linearly from 2 fg to 10 pg at an *R*^2^ value of 0.9661. The detection limit for ABTX was 1.4 amol (1.8 fg, S/N = 5).

### 2.2. Affinity of ABTX to Rat Brain Synaptosomes

The affinity of ABTX against the rat synaptosome was determined by the binding assay using tritium-labeled BTX (^3^H-PbTx-3) at 1 nM and the synaptosome fraction at 50 µg protein/mL. The direct plot of PbTx-3 and ABTX concentrations against the percentage of maximum values and Hill’s plot for PbTx-3 and ABTX are shown in Figure 5. The high correlation coefficients of the two curves suggest that the binding assay experiments showed a good reliability. The values of IC_50_ of PbTx-3 and ABTX were 8.8 ± 0.42 nM and 3.8 ± 0.89 nM, respectively. From these data, inhibition constant (*K*_i_) values were calculated as 3.56 ± 0.12 nM for PbTx-3 and 1.66 ± 0.25 nM for ABTX. ABTX possessed a two-fold higher affinity to synaptosomes than PbTx-3. For ^3^H-PbTx-3, the equilibrium dissociation constant (*K*_D_) and the apparent maximum number of the binding sites (*B*_max_) values were determined as 3.53 ± 0.16 nM and 5.67 ± 0.49 pmol toxin/mg protein, respectively, which were used in the calculation of *K*_i_ values of PbTx-3 and ABTX.

### 2.3. Saturation Experiment to Rat Synaptosome by ABTX

The determination of *K*_D_ and *B*_max_ for ABTX and rat synaptosomes was performed by saturation experiments. The experiments were performed with 50 µg protein/mL of synaptosomes. The direct plot of ABTX concentration against chemiluminescence intensity, and a Scatchard plot are shown in Figure 6. The *K*_D_ and *B*_max_ values of ABTX were determined as 0.84 ± 0.03 nM and 6.76 ± 1.11 pmol toxin/mg protein, respectively. These values supported the results shown in the competitive binding assays in Section 2.2. 

### 2.4. Binding Assay Using ABTX for Detection of CTX3C and BTXB2

To examine whether the binding assay using ABTX can detect CTX and BTX or not, we carried out an assay using CTX3C [15] and BTXB2 as the representatives of CTXs and BTXs, respectively. Results show that both CTX3C and BTXB2 were confirmed to inhibit the binding of ABTX to synaptosomes. The direct plots are shown in Figure 7, and the *K*_i_ values of CTX3C and BTXB2 using 3 nM of ABTX were calculated to be 195 ± 22.5 pM and 88.7 ± 19.3 × 10 nM, respectively.

### 2.5. Detection of CTX3C from Fish

The assessment of CTX3C in fish was performed by analyzing samples prepared from three species of fish flesh spiked with CTX3C. In each fish species, a good linearity was observed between the observed and spiked toxin amounts in a range from 0.2 to 1.0 ng CTX3C/g flesh (Figure 8). The theoretical lowest level for spiked CTX3C was 0.2 ng/g flesh (0.2 ppb). Obviously, the interference due to non-specific binding shown in Figure 8 has to be reduced by setting a proper clean-up method. Nevertheless, the high sensitivity and accuracy suggested by the linearity of the dose response, the ease of performance, and finally the low cost can be taken as the promising aspects of the proposed method.

### 2.6. Detection of BTXB4 from Mussels

Detection of BTXB4 was performed on shellfish samples spiked with BTXB4. Results are shown in Figure 9. BTXB4 could be detected at expected levels. Significant interference from the shellfish matrix was observed at the concentration of 0.1 μg/g flesh. Thus, the proposed detection method can be used to detect BTXB4 above 0.20 μg/g flesh. In addition, it possesses enough sensitivity and accuracy to replace the mouse assays. The interference from the matrix was less serious compared with fish.

## 3. Discussion

In CFP and NSP, toxin quantification by ELISA or LC-MS methods is hindered by the large number of the involved toxins, the wide variability of toxin structures and potencies, and the unavailability of reference toxins. Alternatively, functional assays have been developed for the detection of CTXs and BTXs. Taking advantage of CTXs and BTXs to bind and enhance Navch, cytotoxicity assays were developed using neuroblastoma cells. Although highly sensitive, the method requires 18 to 24 h to complete and produce relatively large fluctuations in the results. A competitive binding assay using ^3^H-PbTx-3 has been successful. In view of the inconvenience of the strictly regulated radioactive reagent associated with ^3^H-PbTx-3, we replaced the radioactive ligand with a chemiluminescent reagent. The promising aspect of ABTX for use in binding assays to detect both CTXs and BTXs was revealed. ABTX possessed a higher affinity and a higher number of receptor sites against Navchs compared to ^3^H-PbTx-3.

Even though the proposed method possesses has advantages, there are issues that need further investigation. First, the observed toxicities of BTXB4 in mussel extracts were always underestimated. A similar disagreement observed with CTX3C is probably caused by yellow or red-colored impurities extracted from fish. This disagreement of the toxicity between the expected and observed values was also reported in the case of the animal tissues [16]. From the thin layer chromatography (TLC) of parrot fish extract (data not shown), the spot of high polar substance reminiscent of phosphatidylcholine was observed. Thus, these high-polar lipids are the cause of the interference with the binding of the ligand and Navchs. The non-specific binding to the synaptosome would be reduced by the heat denaturation or the addition of albumin protein. Secondly, the throughput of the assay method is extremely low due to the use of a single cell for the detection of chemiluminescent intensity. The application of a 96-well filter plate, flurorescent ligand (BODIPY^®^-PbTx-2), and multi-photolabel counter may improve this inconvenience [17,18]. Thirdly, the preparation of the chemiluminescent derivative is performed only from BTXB2, which was only obtained from specific shellfish collected in a specific area and date. BTXB, one of the congeners of BTXs that possesses an aldehyde moiety in its molecule, can be used as the starting material. The aldehyde moiety can be used as such for coupling, or it can be elongated with aliphatic diamines to produce a molecule having a primary terminal amine.

The proposed new receptor-binding assay revealed the prospect of ABTX for the detection of CTX3C and BTXB4. This methodology may be a promising detection method for CFP and NSP due to its facileness and preciseness. Although some issues remain, they should be improved by further studies. Finally, this method can be adapted to the other Navch binding toxins, such as tetrodotoxins and saxitoxins.

## 4. Materials and Methods 

### 4.1. Materials

CTX3C was isolated from cultures of *G. toxicus* according to the methods described previously [15]. BTXB2 and BTXB4 were isolated from green-shelled mussels, *Perna canaliclus*, collected in New Zealand [11,12]. PbTx-3 and ^3^H-PbTx-3 were purchased from Chiral Corp (Miami, FL, USA). Rat brains were purchased from Rockland (Sprague–Dawley or Wistar, age and sex mixed, Gilbertsville, PA, USA) or Clea Japan (SD, 8 weeks, male, Tokyo, Japan). A non-toxic parrot fish (*Scarus gibbus*) and two sea basses (*Plectropomus leopardus* and *Epinephelus microdon*) were purchased at a public market in Naha, Okinawa, Japan. Blue mussels (*Mytilus edulis*) collected at Okirai Bay, Iwate, Japan were a generous gift from Kitazato University. All the other reagents and solvents were of analytical grade and purchased from Fujifilm Wako Pure Chemical Corporation (Osaka, Japan) or Kanto Chemicals Co. Inc. (Tokyo, Japan).

### 4.2. Preparation of ABTX

The preparation of ABTX was performed according to the method formerly reported for proteins and peptides with some modifications [19]. Acridinium-I (Dojindo, Kumamoto, Japan) was dissolved in dimethylformamide to give a concentration of 10 mg/mL. BTXB2 (200 µg, 0.19 µmol), Acridinium-I solution (100 µL), and 0.1 M phosphate buffer (pH 8.0, 100 µL) were mixed in a glass microtube at room temperature. The progress of the reaction was checked by TLC (Merck Kieselgel 60, chloroform/methanol/water 12/6/1 (*v*/*v*/*v*), sulfuric acid/methanol (1:1) solution was sprayed and then heated for detection). The reaction was maintained for 3 h until the spot of BTXB2 had disappeared. The reaction mixture was subjected to HPLC purification under the following conditions: column, Shiseido Capcellpak C18 UG (I.D. 4.6 × 250 mm); mobile phase, acetonitrile/10 mM phosphate buffer (pH 5.0) (1:1, *v*/*v*); flow rate, 0.5 mL/min.; detection, Diode Array Detector L-4200 (Hitachi, Tokyo, Japan). Fractions containing ABTX were combined, and the solvent was evaporated under reduced pressure. The residue was dissolved into water and subjected to the desalting procedure using same HPLC system as above using acetonitrile/water (7:3) as an eluent to obtain pure ABTX (59 µg, 22% yield). The structure of ABTX was confirmed by positive ion CID-FAB MS/MS experiments with an HX110/HX110 tandem mass spectrometer (JEOL, Akishima, Japan). The quantification of ABTX was performed by comparing the UV absorbance at 260 nm with that of a standard Acridinium-I solution on a UV-1600 spectrophotometer (Shimadzu, Kyoto, Japan).

### 4.3. Measurement of Chemiluminescent Intensity

Chemiluminescent intensity was measured on a chemiluminescence detector CLC-110 (wave length: 350–650 nm, Tohoku Electronic, Sendai, Japan) assembled with a chemiluminescence counter CLC-10 (Tohoku Electronic, Sendai, Japan) and an open-cell type sample chamber TLU-17 (Tohoku Electronic, Sendai, Japan). CLA/Data Acquisition Software (Tohoku Electronic, Sendai, Japan) was used for data analyses. A sample suspended in the washing buffer, consisting of 5 mM Tris-Hepes buffer (pH 7.4), 163 mM choline chloride, 5.4 mM potassium chloride, 0.8 mM magnesium sulfate, and 1 mg/mL BSA was transferred into a stainless petri dish, and 2 mL of 10 mM phosphate buffer (pH 5.0) was added. The petri dish was placed in the sample chamber and incubated at 40 °C for 100 sec to check the stability of the background counts. A trigger solution consisting of 0.5 N sodium hydroxide/30% hydrogen peroxide (99.5:0.5, *v*/*v*) (0.5 mL) was added with a plastic syringe, and photon counting was started immediately. Chemiluminescent intensity was determined as the integration of photon counts for 50 s after the injection of the trigger solution. During the measurements, the gating time was set at 1 sec.

### 4.4. Preparation of a Synaptosome Fraction from Rat Brain

Rat brains were thawed on ice and suspended in 10-fold (*v*/*w*) buffer-1 consisting of 15 mM Tris-HCl buffer, 0.32 M sucrose, and four protease inhibitors (0.1 mM phenylmethylsulfonyl fluoride, 1 mM iodoacetamide, 1 mM 1,10-phenanthroline monohydrate, and 1 µM pepstatin A). The suspension was homogenized gently with loose-fitting Potter type glass-Teflon homogenizer for 10 strokes. The homogenate was centrifuged at 700× *g* for 10 min, and the supernatant was taken for further purification. Another 10-fold (in *v*/*w*) of buffer-1 was added to the precipitate, and the suspension was centrifuged at 700× *g* for 10 min. The two supernatants were combined and centrifuged at 11,500× *g* for 20 min. After removal of the supernatant, the precipitates were washed twice with 10 volumes of buffer-2, consisting of 50 mM Tris-HCl buffer (pH 7.4), 1 mM EDTA 2Na, and four protease inhibitors by repeating the suspension and the centrifugation at 11,500× *g* for 20 min. Throughout the whole manipulation, the temperature was maintained at 4°C. The resulted precipitate was suspended in buffer-2 and stored at −85°C until use. The protein concentration of the synaptosome was quantified with protein assay kit (BIO-RAD, Richmond, CA, USA) with bovine serum albumin as a standard.

### 4.5. Binding Assay Using ^3^H-PbTx-3 for Evaluating the Affinity of ABTX against Rat Brain Synaptosome

Binding assay using ^3^H-PbTx-3 was performed with the protocol reported previously [20]. In 8-mL disposable test tubes, 0.5 mL of ^3^H-PbTx-3 solution in incubation buffer (final 1 nM), consisting of 50 mM Tris-Hepes buffer (pH 7.4), 130 mM choline chloride, 5.5 mM glucose, 0.8 mM magnesium sulfate, 5.4 mM potassium chloride, 1 mg/mL BSA, and 0.01% (*v*/*v*) polyoxyethylene-10-tridecylether was mixed with the appropriate volume of sample solution in methanol and 0.5 mL of the synaptosome solution (final 50 μg protein/mL). After incubating for 2 h at 4 °C, 1 mL of ice-cold washing buffer was added to each tube to stop the reaction. The solution was rapidly filtered through a GF/B glass-fiber filter (Whatman, Kent, UK), which was previously soaked in 0.1% polyethyleneimine for more than 2 h to minimize the non-specific binding of ^3^H-PbTx-3 to the filter, with a cell harvester M-12R (Brandel, Gaithersburg, MD, USA). Then, the residue on the glass-fiber filter was washed with 4 mL of ice-cold washing buffer three times. The glass-fiber filter was placed into the scintillation counting vial, and 8 mL of scintillation cocktail EX-H (Dojindo, Kumamoto, Japan) was added. After mixing the vial vigorously, the radioactivity was measured with β-counter 1900TR (Packard, Meriden, CT, USA). The affinity of ABTX was compared with PbTx-3, and the inhibition constants (*K*_i_) of PbTX-3 and ABTX were determined from the IC_50_ of these compounds using the following equation.
*K*_i_ = (IC_50_ − Rt/2)/(1 + [L]/*K*_D_)
Rt: Total receptor concentration; [L]: 3H-PbTx-3 concentration; *K*_D_: Dissociation constant of 3H-PbTx-3 determined by saturation experiment

### 4.6. Binding Assay Using ABTX for the Detection of CTX or BTX

The binding assay using ABTX was performed as follows. The two buffers used in this experiment—incubation and washing buffers—were the same as in the case of the binding assay using ^3^H-PbTx-3. In a 1.5-mL microtube, 0.5 mL of ABTX dissolved in the incubation buffer (final 3 nM) was mixed with an appropriate volume of sample solution in methanol, and 0.5 mL of synaptosome suspension in the incubation buffer (final 50 μg protein/mL). The tube was kept at 0 °C for 2 h with gentle shaking, and then centrifuged at 13,000× *g* for 2 min (0 °C). From the supernatant, 0.9 mL of buffer was gently removed, and the precipitates were re-suspended in another 0.9 mL of ice-cold washing buffer, and centrifuged at 13,000× *g* for 2 min (0 °C). This washing manipulation was repeated, and the precipitate was suspended in 0.2 mL of the washing buffer. The suspension was transferred into a stainless Petri dish cell for chemiluminescence detection.

### 4.7. Assay of CTX3C in Fish Flesh Using ABTX

Based on the previous purification method for CTX3C [15], test samples were prepared from fish flesh using two kinds of Sep-Pak cartridges (Waters, Milford, MA, USA) (Figure 10). The flesh of *S. gibbus* (150 g) was homogenized, and 450 mL of acetone was added. The suspension was filtered, and the residue was again extracted with the same volume of acetone. The filtrate was combined, and the solvent was evaporated under reduced pressure. The residue was dissolved into 80% methanol (150 mL) and partitioned with hexane (150 mL). The 80% methanol layer was again partitioned with hexane (150 mL), and two 80% methanol layers were combined and evaporated under reduced pressure. The residue was dissolved into 95% methanol and subjected to Sep-pak plus PS-1. The column was washed with 95% methanol and eluted with 80% 1-propanol. The eluent was evaporated and dissolved into hexane/acetone (4:1) and subjected to Sep-pak Florisil. The column was washed with hexane/acetone (4:1) and eluted with acetone/methanol (9:1). For *P. leopardus* and *E. microdon*, 100 g of flesh was used, and the solvents were reduced to 2/3 according to the weight of the flesh. CTX3C was added to the acetone/methanol (9:1) fraction obtained from the flesh of *S. gibbus* to make 0.2, 0.3, 0.5 and 1.0 ng/g flesh and to *P. leopardus* and *E. microdon* to make 0.3 and 1.0 ng/g flesh. The fish samples were confirmed to be non-toxic by measuring the chemiluminescent intensity of the fish flesh fraction equivalent to 10 g of flesh that showed below 10% of the intensity observed in the control group (<0.002 MU/g flesh). Each sample was tested at three different concentrations to give a 50% binding inhibition (IC_50_) value. The toxicities of the tested samples were calculated from the 50% binding inhibition value of the CTX3C standard.

### 4.8. Assay of BTXB4 in Blue Mussels Using ABTX

Test samples were prepared according to the purification method for BTXB4 [12] with modifications (Figure 11). The hepatopancreas of blue mussel was extracted twice with acetone, and the combined extract was evaporated to dryness. The residue was transferred in a separation funnel with diethyl ether (3 mL × 2) and water (3 mL). The diethyl ether layer was evaporated under the gentle flow of nitrogen gas, and the residue was partitioned between 80% methanol (3 mL) and hexane (3 mL × 2). The 80% methanol fraction was subjected to a Sep-pak PS-I cartridge, and substances retained on the cartridge were eluted with 1-propanol (5 mL). BTXB4 was added to the 1-propanol fraction to make 100, 200, 400, and 800 ng/g shellfish. The shellfish samples were confirmed to be non-toxic by measuring the chemiluminescent intensity of the shellfish fraction equivalent to 80 mg of flesh, which showed below 10% of the intensity observed in the control group (<0.063 MU/g flesh). The binding assay was performed as described for CTX3C.

## Figures and Tables

**Figure 1 toxins-11-00580-f001:**
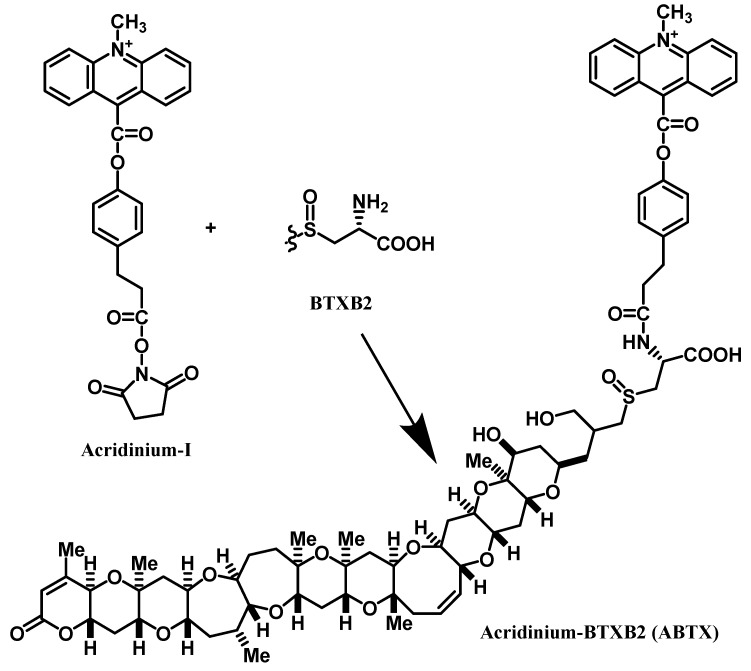
Preparation of acridinium-BTXB2 (ABTX).

**Figure 2 toxins-11-00580-f002:**
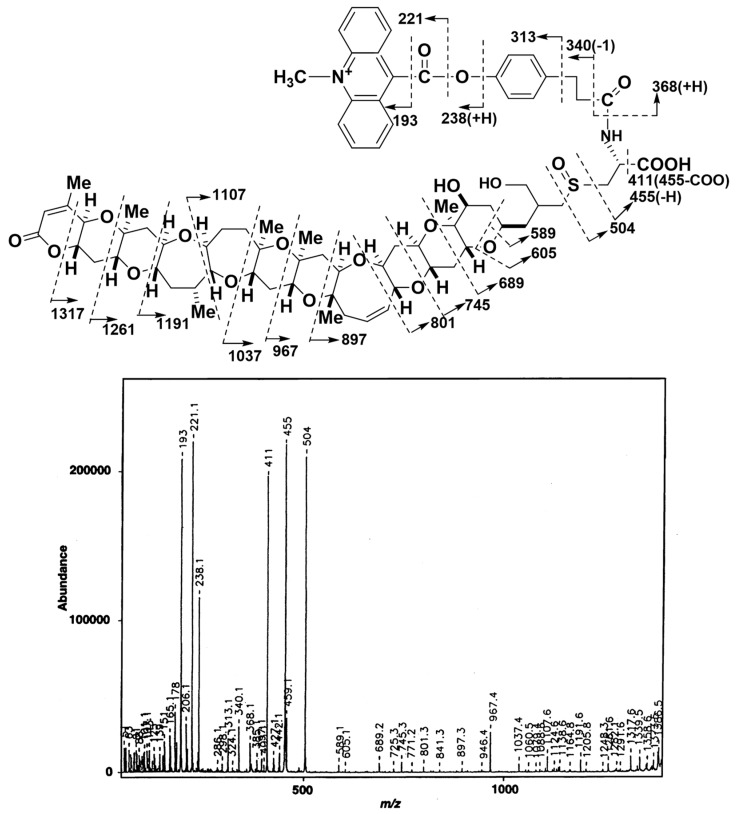
FAB-CID-MS/MS spectrum of ABTX (positive mode, precursor ion; [M+H]^+^ = *m/z* 1401, matrix; 2,2’-dithiodiethanol). FAB: fast atom bombardment, CID: collision-induced dissociation, ABTX: acridinium BTXB2.

**Figure 3 toxins-11-00580-f003:**
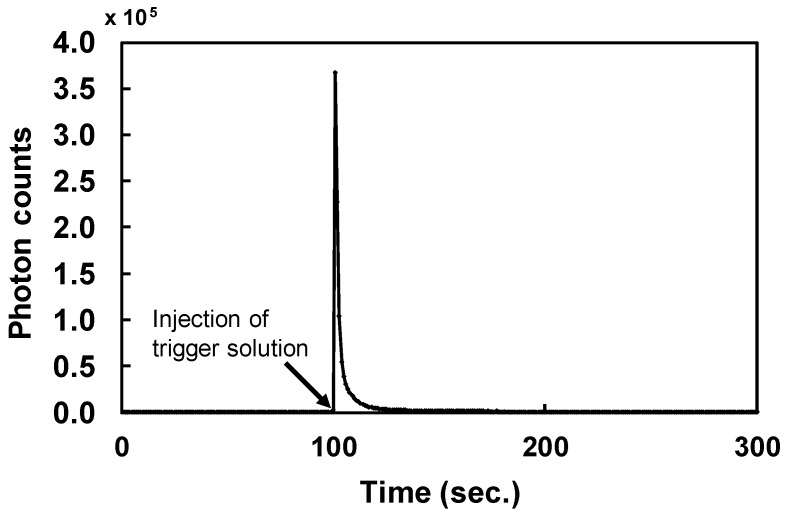
Chemiluminescent profile of ABTX (1 pg).

**Figure 4 toxins-11-00580-f004:**
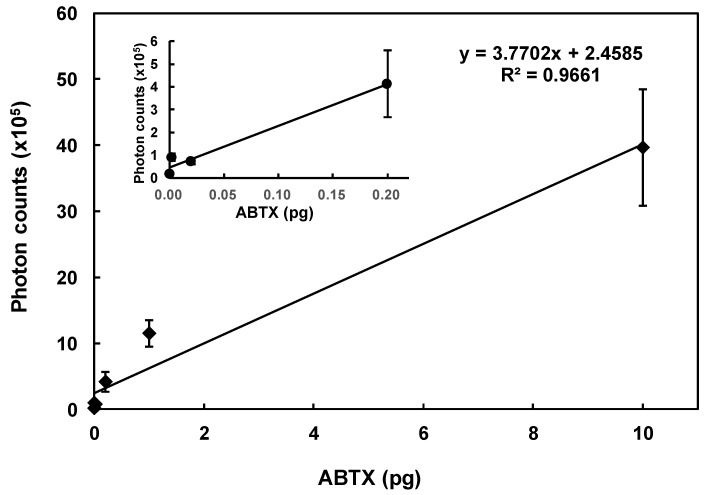
Linearity of ABTX detection: the inserted graph shows the enlargement from 0.00 to 0.20 pg of ABTX, symbols show the means, and error bars show the differences of duplicates.

**Figure 5 toxins-11-00580-f005:**
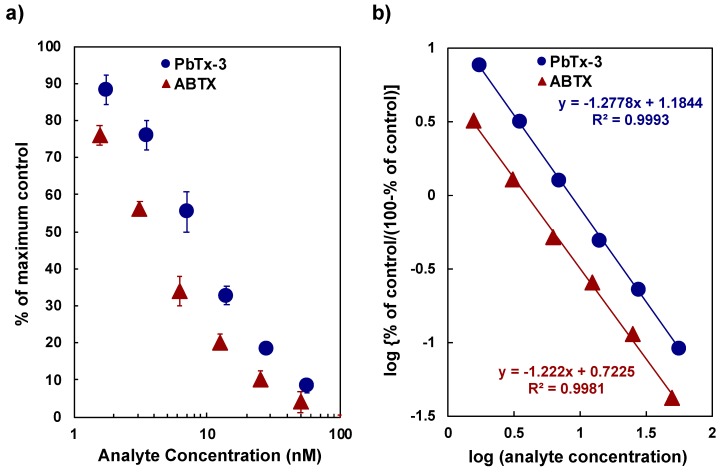
Competitive binding assay between ^3^H-PbTx and PbTx-3 or ABTX: (**a**) direct plot, (**b**) Hill’s plot.

**Figure 6 toxins-11-00580-f006:**
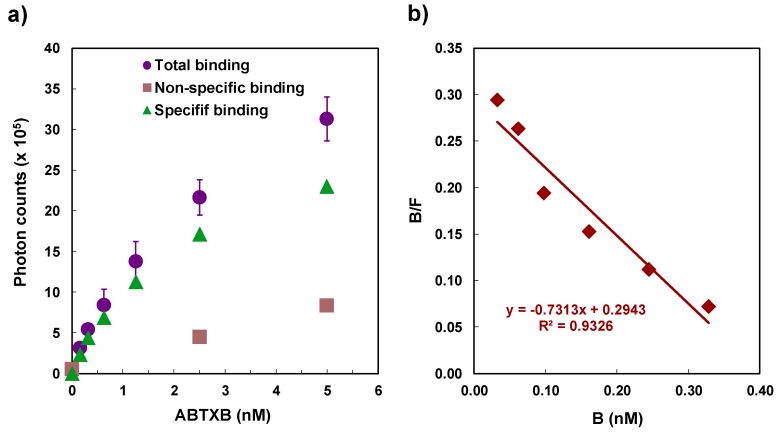
Saturation experiment of ABTX against rat brain synaptosome: (**a**), direct plot; (**b**), Scatchard plot. Data show the means ± S.D. of triplicates.

**Figure 7 toxins-11-00580-f007:**
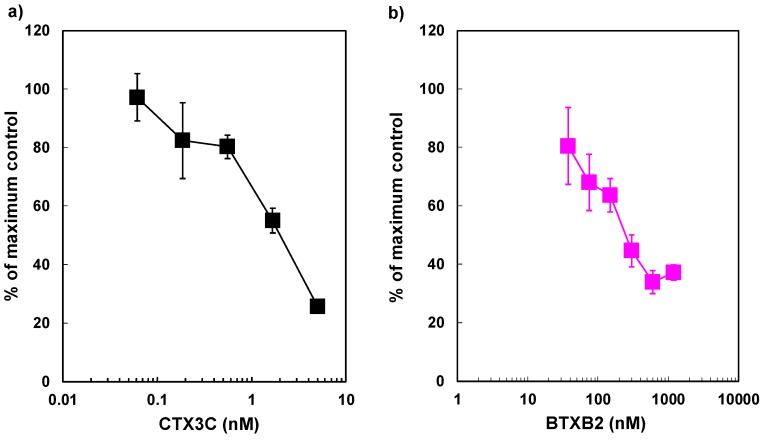
Competitive binding assay between ABTX and (**a**) CTX3C of (**b**) BTXB2. Data show the means ± S.D. of triplicates. CTX: ciguatoxins.

**Figure 8 toxins-11-00580-f008:**
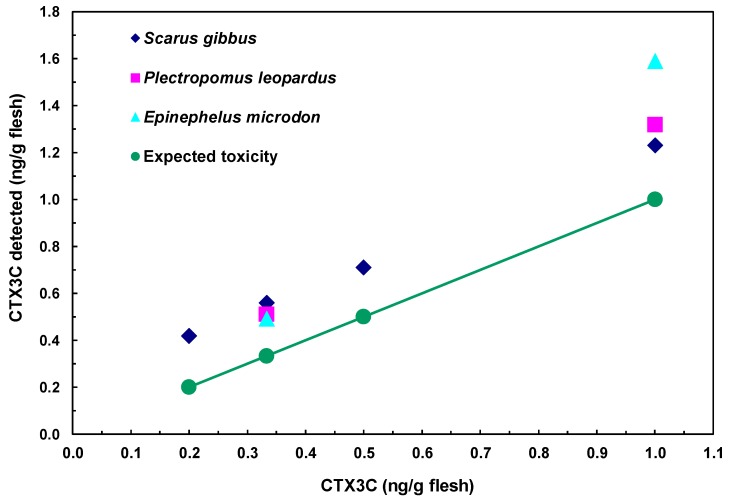
Detection of CTX3C from fish extracts.

**Figure 9 toxins-11-00580-f009:**
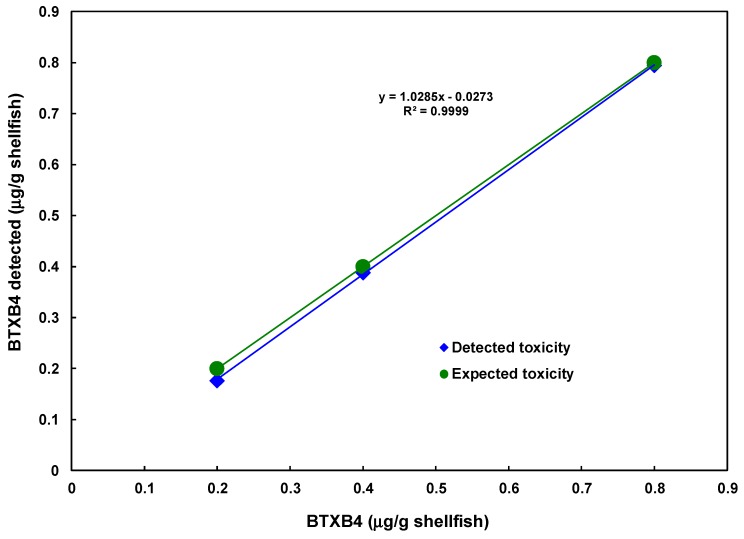
Detection of BTXB4 from mussel extract.

**Figure 10 toxins-11-00580-f010:**
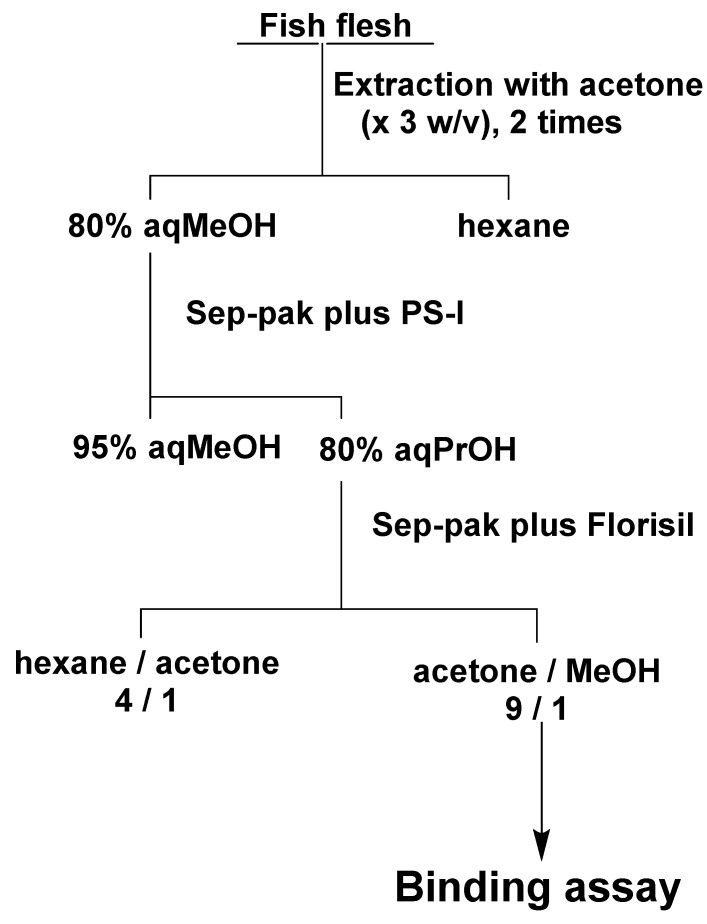
Purification of fish flesh for CTXC3 assay: MeOH, methanol; PrOH, 1-propanol.

**Figure 11 toxins-11-00580-f011:**
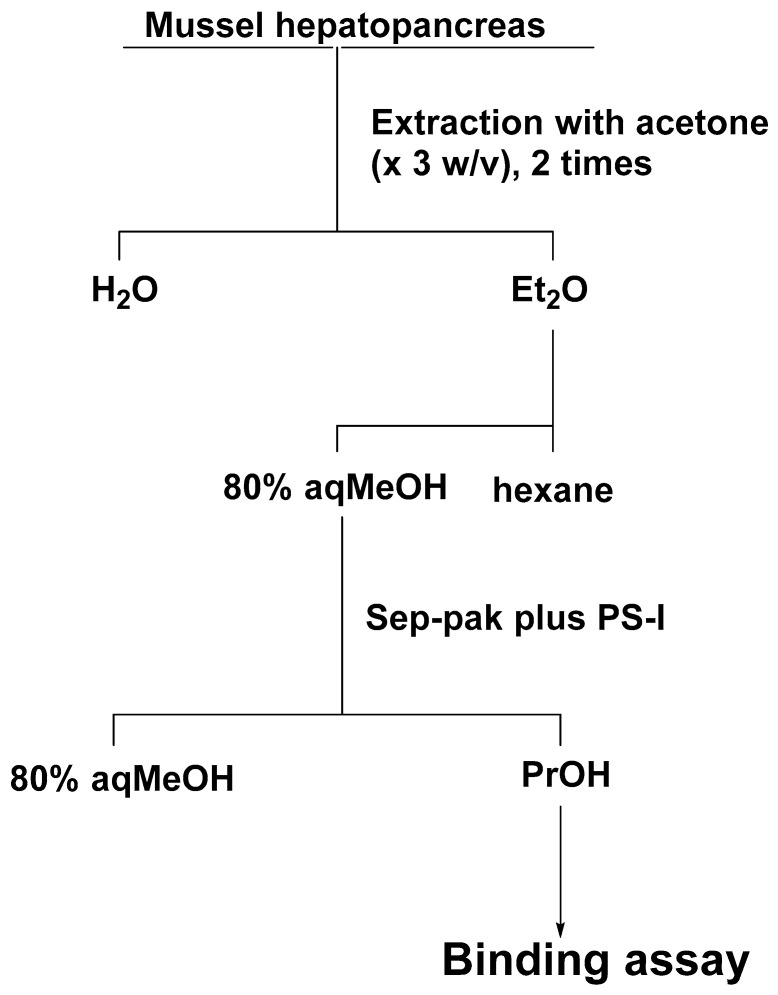
Purification of shellfish flesh for BTXB4 assay: Et_2_O, diethyl ether; MeOH, methanol; PrOH, 1-propanol.

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
