# Peer review of "Chemiluminescent Receptor Binding Assay for Ciguatoxins and Brevetoxins Using Acridinium Brevetoxin-B2"

_toxins, 2019, doi:10.3390/toxins11100580_

Round 1

Reviewer 1 Report

Lines 57&58 - The low yield of the ABTX was attributed to the use of buffer. This doesn't statement doesn't clearly explain why the yield was impacted by the use of buffer or if or how the yield could be improved. I would recommend explaining why you think the yield was low. In other words why did the use of buffer result in a low yield and is it preventable? Presumably it is a pH effect, not a specific buffer effect.   

Figure 4 - The small graph within the graph is not labelled on either axis and there is no title or explanation for what this is.

Figure 5 - I don't understand why two graphs are necessary, they show essentially the same thing. I would suggest reducing this to a single graph or explain in the results what you are trying to show in each. It should state that averages were used on the Hill's plot from a stated number or replicates if you use this graph.

Figure 7 - This has the same caption as figure 6. It appears that one graph a) is for CTX3C and the other b) is for BTXB2. Please fix.

Just a general comment.. I was left wondering why only CTX3C and BTXB4 tested? There was no explanation for this. Are these representative of real samples? It would be interesting to see how CTX1b and other brevetoxins performed. I wonder how naturally contaminated samples that have real profiles would compare between this technique and other analytical methods. 

Author Response

Lines 57&58 - The low yield of the ABTX was attributed to the use of buffer. This doesn't statement doesn't clearly explain why the yield was impacted by the use of buffer or if or how the yield could be improved. I would recommend explaining why you think the yield was low. In other words why did the use of buffer result in a low yield and is it preventable? Presumably it is a pH effect, not a specific buffer effect.

The low yield is due to the use of water which degrades the n-hydroxysuccinimidyloxy moiety. The buffer itself does not affect the yield. The manuscript was corrected as in the right column.

Water was used in order to improve the solubility of BTXB2 into the reaction mixture. BTXB2 is a zwitter ionic molecule which does not dissolve in organic solvents. Thus, the use of water is unavoidable.

Line 61;The low yield can be attributed to the use of buffer solution water (buffer solution) in the reaction mixture.

Figure 4 - The small graph within the graph is not labelled on either axis and there is no title or explanation for what this is.

The small graph is a magnified graph of the big graph from 0.00 to 0.20 pg of ABTX. Labels in the axes were added and the description was added to the legend. 

Line 85;Linearity of ABTX detection: the inserted graph shows the enlargement from 0.00 to 0.20 pg of ABTX and symbols show the means and error bars show the differences of duplicates.

Figure 5 - I don't understand why two graphs are necessary, they show essentially the same thing. I would suggest reducing this to a single graph or explain in the results what you are trying to show in each. It should state that averages were used on the Hill's plot from a stated number or replicates if you use this graph.

In Figure 5, a) is a direct plot and b) is a Hill’s plot. Hill’s plot is needed to calculate IC50 values, which are intersection of x axis and the approximate curve. In addition, correlation coefficients of the approximate curve of in the Hill’s plot shows reliability of the experiment. Description were added to the main text as in the right column.

Line 91;

The high correlation coefficients of the two curves suggest that the binding assay experiments showed a good reliability.

Figure 7 - This has the same caption as figure 6. It appears that one graph a) is for CTX3C and the other b) is for BTXB2. Please fix

Figure legend was corrected as the reviewer.

Line 122;

Competitive binding assay between ABTX and a) CTX3C of b) BTXB2.Saturation experiment of ABTX against rat brain synaptosome: (a); direct plot, (b); Scatchard plot.

Just a general comment.. I was left wondering why only CTX3C and BTXB4 tested? There was no explanation for this. Are these representative of real samples? It would be interesting to see how CTX1b and other brevetoxins performed. I wonder how naturally contaminated samples that have real profiles would compare between this technique and other analytical methods.

The reason for using CTX3C for the experiment is CTX3C is the only ciguatoxin available through culture of dinoflagellate. There are other ciguatoxins dominant to ciguatera poisoning. However, their availabilities are very low. For BTXB4, the compound is the major causative toxin for neurotoxic shellfish poisoning. We could not utilize toxic fish or shellfish sample due to the lack of method detecting the toxins accurately. The aim of this report is to demonstrate the accuracy. Thus, the spiked samples were utilized.

Reviewer 2 Report

Keep the good work

Author Response

Thank you for your kind comment.

Reviewer 3 Report

In my opinion, the research presented in Chemiluminescent Receptor Binding Assay for Ciguatoxins and Brevetoxins using Acridinium Brevetoxin-B2 is sound and clearly written. I think that the topic is extremely interesting and it makes me think about whether we can apply techniques like ultrahigh resolution mass spectrometry to identify unknown oxidized congener metabolite products (if there are still some that may not be identified). 

There are a couple small grammar fixes.

L46: metabolite to metabolites

L51: Needs to be reworded to something like, "...a limit of detection as low..."

There are other minor instances like this throughout. 

Author Response

L46: metabolite to metabolites

Agreed and corrected.

L51;metabolite metabolites

L51: Needs to be reworded to something like, "...a limit of detection as low..."

Agreed and corrected.

L55;In addition, the detection limit as low as 1.4 amol of for ABTX was achieved.

There are other minor instances like this throughout.
Agreed and English grammar check was performed by a native speaker of English and the corrections were indicated by yellow highlights and red letters.

Corrections were indicated by yellow highlights and red letters.